# Adherence to Mediterranean Diet and Food Neophobia Occurrence in Children: A Study Carried out in Italy

**DOI:** 10.3390/nu15245078

**Published:** 2023-12-12

**Authors:** Annalisa Di Nucci, Simone Pilloni, Umberto Scognamiglio, Laura Rossi

**Affiliations:** 1Department of Cardiovascular, Endocrine-Metabolic Diseases and Aging, Italian National Institute of Health, 00161 Rome, Italy; annalisa.dinucci@gmail.com; 2CREA Council for Agricultural Research and Economics, Research Centre for Food and Nutrition, 00178 Rome, Italy; pllsmn99@gmail.com (S.P.); umberto.scognamiglio@crea.gov.it (U.S.)

**Keywords:** Food Neophobia, adherence to Mediterranean Diet, sociodemographic characteristics, children, Italy

## Abstract

Food Neophobia (FN), defined as the reluctance to eat new or unfamiliar foods, mainly concerns fruit, vegetables, and legumes, typical of the Mediterranean Diet (MD). Considering these premises, this study aimed to clarify the relationship between FN and AMD in a sample of Italian children and their association with some socio-demographic factors and children’s nutritional status. A sample of 288 children aged 3–11 years participated in an assessment carried out with a questionnaire evaluating FN and AMD, respectively, with the Child Food Neophobia Scale (CFNS) and the KIDMED test. Most of the sample showed an intermediate (67.3%) or high level of FN (18.1%), with high rates among 6–11-year-old children (63.9%) and especially in those who were the only child (50%). The AMD was mostly low (29.5%) or medium (54.8%) and reached lower levels among higher neophobic children (51.9%; *p* value < 0.05). The present results confirm the study hypothesis that FN is a driver of MD abandonment and shows the positive effects on children’s eating habits and siblings. Finally, this study proves the relevance of adopting effective feeding strategies against FN to avoid its maintenance in adulthood and the detrimental effects on future overall health and well-being.

## 1. Introduction

Food preferences in children are linked to exposure to a high diversity of food in the early years, which improves dietary variety at later developmental stages [1]. The reluctance to eat new or unknown foods is defined as Food Neophobia (FN) [2,3]. It is a normal developmental phase, occurring without the distinction of gender, that typically peaks between 2 and 6 years of age and then gradually decreases into adulthood [2,4]. However, FN could have pervasive implications for food-related behaviors impacting the sensory perception of food in the sense that people with high levels of FN reported limited enjoyment of food [5].

Typical foods of the Mediterranean Diet (MD), such as fruit, vegetables, whole grains, and legumes, are related to children’s FN [1]. MD is negatively associated with the risk of non-communicable chronic diseases, such as diabetes, cardiovascular diseases, and all causes of mortality [6,7,8,9,10,11] when compared to the Western Diet [12]. Hence, the nonadherence to MD in developmental ages is a behavior that needs to be corrected. Eating habits and behaviors are shaped in childhood and then maintained in adulthood [13]; hence, it is important to act early on neophobic behavior to ensure that it does not impact diet quality [14].

Over the last years in Mediterranean Countries, a progressive abandoning of MD has been observed either in children or adults [15,16]. Among children in Italy, several reports showed an inadequate consumption of typical foods of MD, such as fruit, vegetables, and legumes [17,18].

Among neophobic children, it is common to find a high consumption of foods rich in saturated fatty acids and sugars [19,20,21], typical of a Western-style diet [12]. Poor dietary variety and quality due to FN could be a predictor of childhood overweight/obesity; however, to date, findings are not univocal [22,23,24,25,26].

FN is primarily a hereditary trait, in which the genetic determinants account for 78% [27]; however, several socio-environmental factors can influence its development [28,29]. The eating patterns and feeding strategies that parents adopt are strongly associated with the development of FN in children [14,30,31]. During family mealtimes, due to the social facilitation mechanism [32], children can observe and acquire the eating habits of their parents and peers (e.g., siblings) [33,34,35,36].

Parents’ low consumption of fruit and vegetables is strongly associated with the limited consumption of these foods by children [37] because of the transposition of parents’ food preferences on children. The consequences of this attitude could be the children’s restricted experience of learning about various types of food as well as a poor variety of dietary preferences [2,28,38]. Consistently, several studies demonstrated that the mother’s high level of neophobia is correlated with the highest neophobia in children [25,39,40,41,42,43]. Low parent education levels could be one of the causes of low Adherence to the Mediterranean Diet (AMD) and the high level of neophobia in children [44,45,46,47]. Moreover, parents’ insufficient nutritional knowledge about what foods are healthy for their children and how to offer them [19] could exacerbate either neophobia or low AMD.

To the best of our knowledge, few studies have evaluated the relationship between FN and AMD in the pediatric population, mostly conducted in Spain [1,48]. For these reasons, the present study aimed to clarify the correlation between FN and AMD in children in an Italian sample. The specific purposes of this work were the analysis of the socio-demographic factors influencing AMD and FN as well as the relationship between AMD and FN and children’s nutritional status.

The basis for the hypothesis of this work was that a high level of FN corresponds to worse AMD. Consequently, the research questions this study would like to answer are as follows: (i) Could FN be a driver of AMD? (ii) what child behavior, either in-line or not-in-line with MD, could be identified as most related to FN? (iii) what aspects characterizing the family influence children’s neophobia?

## 2. Materials and Methods

### 2.1. Study Design

The present assessment is a cross-sectional study carried out on a sample of children aged 3–11 years. A flow chart visualizing the different phases of the study is presented in Figure 1. The fieldwork lasted 3 months, from 9 March to 11 May 2022. The only inclusion criterion was the class of age. Children’s health status and lifestyle aspects were not used as inclusion/exclusion criteria in consideration of the fact that FN influences food habits and behaviors, nutrient intake, and, in some cases, anthropometric parameters and is less related to other conditions [49,50,51].

The sample size was calculated according to Pourhoseingholi et al. [52] using the following formula: This is example 1 of an equation:n = [Z^2^ × P(1 − P)]/[d^2^] (1)
where n is the sample size, the Z value is 1.96, corresponding to a 95% level of confidence, and P denotes the phenomenon to be measured. In this case, P is the prevalence of FN in children, which was estimated at 25%, according to Predieri et al. [53], and expressed as a decimal. Finally, d is the precision level of 5%, expressed as a decimal. The resulting sample size was 288 subjects.

Most of the sample (227 subjects) resulted from direct data collection carried out by randomly interviewing the selected parents of children attending schools and parks in Rome’s fifth municipality and in the surrounding area. The remaining sample (61 subjects) was obtained through an online administration of the questionnaire distributed through instant messaging apps (e.g., WhatsApp) and the social platforms of Instagram and Facebook.

### 2.2. Data Collection Procedure

Following the European Commission General Data Protection Regulation (679/2016), those willing to participate signed a privacy policy and consent form concerning the collection and processing of socio-demographic data in advance. Before starting the data collection, participants were informed about the objective of the research, the consequent statistical analysis, and the intention to publish the results of the assessment in a scientific journal. Participation in the study was fully voluntary and anonymous, and subjects could withdraw from the study at any time for any reason. This study was conducted according to the guidelines of the Declaration of Helsinki [54]. The present research is not considered either medical experimentation or a direct intervention on human subjects with diet changes or formulated food administration and did not involve any invasive procedures. In addition, the Council for Research, Economics, and Agriculture (CREA), which performed the study, is part of the National Statistical System (SISTAN) and guarantees individual data protection [55]. Hence, an additional ethical committee review of the study protocol was considered unnecessary once informed consent was obtained.

### 2.3. The Questionnaire

The questionnaire was developed according to the objective of the study and was completed by an adult who acted as a caregiver of the assessed child. It consisted of three sections reported in the Appendix A. The first section included questions on sociodemographic characteristics (the gender, age, weight, and height of the child, ethnic group of the family, household income, number of workers in the family, and number of children in the family). Weight status identification was based on a Body Mass Index (BMI) comparison with the growth charts of the WHO and their cut-offs [56,57].

The second section investigated FN behavior in children with the Child Food Neophobia Scale (CFNS) developed by Pliner [58] and validated in Italy by Laureati et al. [59]. The CFNS consists of ten items (five referring to neophiliac and five to neophobic attitude) evaluated with a 7-point scale ranging from 1 = “I strongly disagree” to 7 = “I strongly agree”.

The individual CFNS scores were computed according to Predieri et al. [53] as the sum of the scores of the 10 items, reversing the neophiliac items to ensure that the univocal sense of all the responses was obtained. Therefore, the scores theoretically ranged from 10 to 70, with higher scores reflecting higher FN levels. Three groups of individuals were identified according to the following calculation: neutrals (score in the interval mean ± 1SD), neophobic (score > mean + 1SD), and neophiliac (score < mean − 1SD) [20,26,49].

The third section of the questionnaire assessed the AMD in children with the “Mediterranean diet and quality index in children and adolescents KIDMED Test” [60], a questionnaire specifically developed to evaluate if children’s eating habits are based on principles of the MD pattern. The test consists of 16 questions with closed answers (yes or no). The 12 questions denoting attitude in line with the MD principle were assigned a value of +1 (e.g., uses olive oil at home); the 4 questions corresponding to attitude not in line with the MD principle were scored −1 (e.g., skipping breakfast). The KIDMED score ranges from 0 to 12, and the sums of the values from the administered test are classified into three levels of AMD: ≥8 indicates “optimal” AMD; 4–7 points indicate “average” AMD (improvement needed); ≤3 points indicate “very low” AMD.

Cronbach’s Alpha was calculated for the combined 26 items of CFNS and KIDMED, resulting in good (α = 0.80) internal consistency and the good reliability of the scale measured [61].

### 2.4. Statical Analysis

Before starting the statistical analysis, the data set was checked. The data collection was managed by one of the principal investigators of the study without the use of data collectors. Therefore, no incomplete questionnaires, incongruences, typing errors, or outliers were detected, and no elimination of the units was necessary. Descriptive statistics of the data collected were produced. Single continuous variables, assessed for normality using the Kolmogorov–Smirnov test, were presented as the mean and standard deviation. The other variables were categorized and presented as a percentage (%). A contingency analysis was performed to check the associations between variables. Specifically, double-entry tables were processed, and the Chi-square test of independence was applied, along with post hoc tests to check pairwise comparisons, with Bonferroni corrections of the *p*-values and the Cramér’s V to estimate its effect size. The test of independence on the mean was applied to compare continuous variables with categorical variables. Spearman’s rank correlation coefficient (Spearman’s r) was calculated for the comparison between ordinal variables.

Linear regressions were carried out to investigate the relationship between FN, both as a score and as a level, and AMD. To check the effect of potential confounders on this association, linear regressions adjusted for all socio-demographic variables were carried out. The results were considered significant for *p*-value < 0.05.

Statistical analysis was performed using Microsoft^®^ Excel and R version 4.3.2 (updated on 31 October 2023) software.

## 3. Results

### 3.1. Characteristics of the Sample

Table 1 shows the sociodemographic characteristics of the sample. Females accounted for 42.7% and males 57.3%; a large part of the sample were Caucasians (79.8%), and even a certain level of multi-ethnic origin was observed (12.9% Eurasians, 4.5% Africans, 1.4% Hispanics, and 1.4% Asiatic). Most of the assessed children (71.2%) were aged 6–11 years. Consistently, 65.6% of the sample attended primary school. In most cases, both parents were employed (87.8%). Consistently, the household income for most families ranged between 25,000–40,000 euros (35.8%) and 10,000–25,000 euros (31.6%).

A comparison of BMI with growth charts of the WHO and their cut-offs [56,57] showed that almost half of the sample (44.8%) was of normal weight, while 20.8% and 28.8% of respondents were respectively overweight or obese, and only 5.6% of the population resulted in an underweight score, respectively.

### 3.2. The Adherence to Mediterranean Diet Assessment (KIDMED Test)

According to the categorization of KIDMED scores, 29.5% of respondents had a low AMD, 54.8% showed an average AMD, and only 15.6% resulted in a high AMD (Figure 2). The inadequate consumption of fruit (<2 servings/day: 61.8%), vegetables (<2 servings/day: 73.3%), legumes (<1 serving/week: 53.8%), nuts (<2/3 servings/week: 74.7%), and fish (<2/3 servings/week: 34.7%) was observed; on the other hand, the excessive consumption of sweets (>1 serving/day: 39.9%) and fast food (>1 time/week: 18.8%) was reported. Among the habits in line with MD, it should be reported that the occurrence of breakfast (88.2%), mainly with dairy products (91.7%) and commercially baked goods and pastries (81.6%), was clear. The consumption of cereals or grain (e.g., bread) for breakfast was less common (18.8%) (Appendix A).

In males, AMD is more common than in females (65.2% vs. 34.8% *p*-value < 0.05) at a moderate magnitude of the association (Cramer’s V 0.18). The occurrence of having both parents employed was associated with AMD in children (93% vs. 7% *p*-value < 0.05) at a low/moderate magnitude of the association (Cramer’s V 0.18). Having siblings was associated with AMD (*p*-value < 0.05) at a moderate magnitude (Cramer’s V 0.13) and with a significant correlation (Spearman’s r 0.10; *p*-value < 0.05). For the other sociodemographic characteristics, no significant association with AMD was found (age: dependency ratio on average 0.004). Weight status was not significantly associated with AMD; however, higher levels of AMD were found in normal-weight children (42.2%) than in overweight (24.4%) or obese (24.4%) individuals (Table 2).

### 3.3. Food Neophobia Assessment

The average CFNS score was 42.2 (SD = 14.04), resulting in a large majority of the sample having an intermediate level of FN (67.3%) followed by a high (18.1%) and low (14.6%) level (Figure 3).

Having siblings influenced the occurrence of FN in the sense that among neophiliac children, a lower proportion of those who were an only-child was found with respect to the neophobic counterparts (16.7% vs. 50%; *p* value < 0.05), confirmed by a moderate Cramer’s V (0.16). This inverse relationship was confirmed by Spearman’s significant correlation (r = −0.21; *p*-value < 0.05). FN did not show significant associations with socio-demographic variables (age: dependency average of ratio 0.01). The relationship between FN and weight status was also studied, resulting in a significant association. Higher rates of normal weight than overweight/obesity were found among the respondents with lower levels of neophobia (61.9% vs. 38.1%; *p* value < 0.05). However, even in higher neophobic individuals, about half were normal weight (42.3%) (Table 3). A strong Cramer’s V (0.22) was observed, while a non-significant Spearman’s rank correlation was found.

### 3.4. The Relationship between Adherence to the Mediterranean Diet and Food Neophobia

To test the study hypothesis that a high level of FN could be a driver of poor AMD, the association between these variables was assessed, and significant results were observed (*p*-value < 0.05). Specifically, almost half of children with a high level of FN (51.9%) showed poor AMD (Table 4). Cramer’s V showed a strong effect size (0.25), and this significant correlation suggested a highly inverse relationship (Spearman’s r = −0.51; *p*-value < 0.05).

To assess the intake of different food groups in both neophobic and neophiliac children, the association between FN and the items of the KIDMED test was carried out. In lower neophobic children, adequate consumption of fruit (2 servings/day; 76.2%), vegetables (2 servings/day; 66.7%), and legumes (>1 serving/week; 92.9%) was found (*p*-value < 0.05). Otherwise, in higher neophobic respondents, insufficient consumption of fish (<2–3 servings/week; 48.1%; *p*-value < 0.05) was observed. On the other hand, neophiliac children were less frequent consumers (<once a week) of fast food than neophobic respondents (97.6% vs. 75%; *p* value < 0.05). Having breakfast did not show a significant association with FN in almost all both neophiliac (95.2%) and neophobic (80.8%) children who did not skip breakfast. Concerning food choices, the consumption of milk and commercially baked goods and pastries was not related to FN; instead, consuming cereals/grains for breakfast was higher, even nonsignificant, in participants with a low level of FN than in those with a high level (31% vs. 11.5%) (Table 4).

Among higher neophobic children, the lowest daily fruit and vegetable (F and V) consumption frequencies were observed. In particular, 80.4% of the children with an intermediate level of FN and 65.4% with a high level of FN consumed one fruit or fruit juice every day. The consumption of two servings per day was reported by 35% of children with intermediate FN and 21.5% of children with a high FN. Similarly, 68% of the children with an intermediate level of FN and 46.1% with a high level of FN consumed fresh or cooked vegetables once a day. These percentages were reduced to 22.7% and 9.6% for two servings per day, respectively. A decreasing trend of weekly fish and legume consumption between the different FN levels was also observed: among higher neophobic children, only 19.2% and 51.9% consumed were, respectively, legumes more than once a week and fish at least 2–3 times per week, contrary to 92.9% and 71.4%, respectively, for legume and fish consumption in their neophiliac counterparts, showing an adequate intake (Figure 4). The strongest effect sizes were found for legumes (Cramer’s V 0.43) with two servings of vegetables (Cramer’s V 0.39) and fruits (Cramer’s V 0.34).

Linear regressions were carried out to investigate the relationship between FN and AMD (Table 5). A significant regressor was found (β = −0.09; *p*-value < 0.05), meaning that the increase in the FN score corresponded to a decrease in the ADM score. Thus, a medium–high ADM was estimated for respondents with a low FN level (ADM score 6.9; *p*-value < 0.05), a medium–low ADM for respondents with a medium FN level (ADM score 4.9; *p*-value < 0.05) and a low ADM for respondents with a high FN level (ADM score 3.5; *p*-value < 0.05).

The linear regressions between FN and AMD, adjusted for all socio-demographic variables, confirmed the above-reported significant association (Table 6 and Table 7). A significant regressor for the FN score was found (β = −0.09; *p*-value < 0.05), and the low FN levels corresponded to high ADM (Low FN, ADM score = 7.1; medium FN, ADM score = 4.9; high FN, ADM score = 3.7).

## 4. Discussion

The present work clarifies the relationship between FN and AMD in a sample of Italian children and describes the association between some sociodemographic factors and children’s nutritional status.

The key finding in the current study highlights that a high level of FN significantly influences the lack of AMD in the assessed sample of Italian children. Notably, more than half of children with elevated FN demonstrated a low AMD. The inverse association between FN and AMD was also confirmed after adjusting for potential confounders. The negative correlation between FN and AMD was reported in other studies carried out in the pediatric population [1,48] as well as in adults [53]. The FN in the present study was a significant barrier to a balanced and healthy diet, and neophobic children resulted in a diet with lower nutritional quality than non-neophobic children.

FN was largely diffused in the present sample since more than half of the respondents showed an intermediate level of FN (67.3%), followed by a high level (18.1%), confirming the rates observed in other Italian studies [24,26,62]. Children exhibiting a high level of FN are hesitant to experiment with a variety of foods, especially those they are unfamiliar with [63]. In the present sample, this resulted in children with a limited eating variety and a dietary behavior far from the principles of MD that could have detrimental effects on future overall health and well-being. Conversely, children who were assessed as less neophobic resulted in general behaviors that were more in line with MD principles.

Foods commonly refused by neophobic children are fruit, vegetables, legumes, and fish, typical of the MD [1]; on the other hand, the Western Diet is characterized by foods such as highly palatable, ultra-processed foods, rich in salt and sugar and refined grains [12] are preferred by neophobic children [19,20,21]. Consequently, FN could be one of the many factors contributing to the shift toward a Western dietary pattern. In suboptimal children, AMD was measured in this study (54.8% average and 29.5% very low), confirming trends of reducing AMD in evolutive ages in Mediterranean areas as resulting in the meta-analysis conducted by Garcia-Cabrera et al. [15] (27% of low AMD among youths ≤ 12 years old).

The observed levels of AMD were related to two socio-demographic variables, namely gender and the level of parental employment. Contrary to other studies [15,47], in this sample, males followed more MD than females. Instead, a higher parental level of employment resulted in a negative factor for AMD as it was associated with higher rates of neophobia in children (93% vs. 80%).

A more detailed examination of the data showed that a large proportion of the respondents did not achieve the recommendations for the consumption of fruit (61.8%), vegetables (73.3%), cereals or grains for breakfast (81.2%), legumes (53.8%) and fish (34.7%) and these included mainly children with an intermediate or high level of FN. Furthermore, F and V, legumes, and fish consumption frequencies tended to decrease as the level of FN worsened. Overall, FN seems to influence vegetable consumption more negatively than fruits. The sweet taste of fruit compared to the bitter taste of some vegetables could justify the difference found in this finding. However, the difference between fruit and vegetable consumption could be minor, considering that the KIDMED item on fruit consumption also refers to fruit juices, normally preferred by children [64].

Regarding more palatable foods, the consumption of commercially baked goods and pastries for breakfast (81.6%), sweets (39.9%), and fast food (18.8%) was frequent, especially among children with an intermediate or high level of FN. These findings are consistent with those obtained in studies conducted in other Mediterranean Countries that recognize FN as a driver behind the abandonment of the MD [1,48]. On the other hand, the unhealthy home food environment and low parent AMD could also be a predictor of maintenance of FN in children.

An interesting result that deserves attention is related to the fact that having siblings influences the occurrence of FN, with a lower percentage of those who are the old child being among neophiliac children compared to their neophobic counterparts (16.7% vs. 50%) probably due to the influence of peers on children’s eating habits [48,65,66].

The literature describes a peak of FN between 2 and 6 years of age with a decrease over this age [2,4,67]; in the present study, however, a high prevalence of FN was found in 6–11-year-old children (63.9% of them with an intermediate level of FN). These results indicate the persistence of neophobic behavior during the child’s growth and the possibility of its maintenance in adulthood, as also observed in a recent study on Brazilian children [68]. The adoption of coercive feeding strategies (e.g., pressure to eat, using food as a reward) could be one of the causes of FN persistence [30,31,50,69,70,71], considering that forcing attitude could have not only an immediate positive impact but a long-term negative effect on the development of preferences for healthy foods [72,73]. The prevalence of FN in adulthood in Italy has been confirmed by another study [53] that showed a strong negative association between FN and AMD and suggested that it could be a predictor of adopting an unhealthy dietary pattern and greater metabolic risk. It is widely documented that FN limits the dietary variety and quality as it results in the rejection of healthy foods, both plants (e.g., fruit, vegetables, pulses) and animals (e.g., fish), and the preference for more palatable and high-calorie foods [19]. Considering these premises, it has been hypothesized that FN may contribute to childhood obesity [22].

The present results did not confirm the association between FN and childhood obesity. In fact, among the less neophobic children, a significantly higher percentage of normal weight was observed compared to overweight/obesity (61.9% vs. 38.1%). However, even in higher neophobic respondents, about half were normal weight (42.3%). The relationship between FN and weight status is an open question, as it has been examined in a few studies that have produced conflicting results. Most of these found no association between the variables [23,24,26,74,75], while a smaller number found a positive relationship in the sense that neophobic children tend to be more overweight/obese [25,40]. Further research is needed to clarify this association, but we can speculate that it depends on the home food environment and the feeding practices adopted by parents: the abundance of calory-dense foods, rich in sugars or saturated fatty acids, and offering them to compensate for the rejection of healthy ones, could favor the development of overweight/obesity in neophobic children.

The strengths and limitations of this study are outlined below. An important strength of this study is the fact that it addresses a research gap in Italy in which limited comprehensive assessments simultaneously investigate both FN and children’s AMD. Understanding the interplay between FN and AMD is essential for promoting healthy eating habits among Italian children. Research in this area could reveal strategies to encourage a more balanced and nutritious diet in children, potentially reducing the impact of FN on their food choices. Another strength is the use of validated and largely employed questionnaires, including the KIDMED test [60] and the Neophobia scale [58], that have been designed to yield consistent and reproducible results. This allowed us to place these findings in a broader context and make meaningful comparisons to existing research, enhancing this study’s significance. Another strength of this work was that the sample size calculation fixed the level of precision of estimations, ensuring the detection of meaningful effects and differences and optimizing efforts and resources when considering the difficulties of data collection in studies with children as the target group. This led to more robust statistical tests and analyses, increasing this study’s ability to detect significant effects and relationships, minimizing the risk of unnecessarily involving participants, and collecting data that might not contribute significantly to the study’s objectives.

This study has limitations. A significant weakness of this research lies in its cross-sectional examination of FN’s association with AMD, as well as the consumption of specific foods like fruits, vegetables, and weight outcomes. Consequently, the study design did not permit a causal relationship to be established. Nonetheless, the findings offer valuable groundwork for future investigations, which should explore the causal connections between neophobia and diet quality during the later stages of childhood. Another important limitation of this study is related to the fact that the assessment relied on respondent’s answers to the questionnaire. This methodology has the intrinsic limitation of the response bias consisting of the fact that respondents may provide inaccurate or socially desirable responses. Dietary intake evaluation is particularly influenced by social desirability bias with a tendency to provide consistent responses, which can potentially lead to less precise representations of actual food consumption [23]. In addition, this study relied on adults’ assessments of neophobic behaviors that could be interpreted differently. All these aspects were partially overcome with the use of largely validated questionnaires. Another limitation of this study is related to the fact that the sample includes medium and high socioeconomic groups, limiting the generalizability of results to the broader population, which is not an objective of the present study. Another limitation of this study could be seen in the sample resulting from a combination of online and face-to-face interviews. However, these mixed methodologies were more and more common in the research with the use of questionnaires [76]. The online systems have the advantage of wider access to participants. Nevertheless, in the present study, the online system resulted in a limited capacity to reach the target; hence, the decision to switch to face-to-face interviews had the advantage of creating a link with the respondents and facilitating their willingness to answer. The reasons for the preference for the direct approach in this study were probably related to the target, which was mothers with young children with limited time to dedicate to online instruments. The decision to mix the methodologies was based on the reported equivalence between the two systems and in consideration of the fact that offering both online and face-to-face interviews places the respondents in the best situation, allowing them to select the most convenient option [77].

## 5. Conclusions

This study confirms the widespread prevalence of FN in the pediatric age in Italy, especially in an only child. Contrary to several studies describing a peak of FN between 2 and 6 years of age, present results also showed the maintenance of neophobic behavior in older children. Confirming the study hypothesis, FN is a driver of low AMD, negatively influencing the consumption of typical foods of MD such as fruit, vegetables, legumes, and nuts and contributing to the high intake of sweets and fast food, characteristic of Western dietary patterns. An inadequate home food environment, unhealthy parents’ eating habits, and the adoption of ineffective feeding strategies could cause the maintenance of FN observed, predisposing to the occurrence of NCDs in adulthood. FN is a significant barrier to achieving a balanced and healthy diet, particularly among children. Neophobic children are at risk of consuming a diet with lower nutritional quality, which could impact their health and development. Encouraging neophobic individuals, especially children, to gradually expand their food choices can be a crucial step in promoting better dietary habits and overall well-being [78].

Future research should expand the utilization of the questionnaires for FN and AMD measurements in other settings and for different purposes. School canteens could be a very interesting area of analysis for FN and AMD in consideration of the high level of food waste reported in the Italian school feeding system [79]. The estimation of the level of FN in these students could be used as background information to interpret the reason for the waste. Food Neophobia Scale and the KIDMED test could be used as evaluative tools to measure the impact of nutrition education programs measuring different aspects of food behaviors. The evaluation of FN could be suitable to identify if there is a relevant proportion of children with extreme neophobic attitudes that limit the impact of educational interventions and need targeted actions. AMD could be used as a measure of behavioral changes and the outcomes of nutrition education programs.

## Figures and Tables

**Figure 1 nutrients-15-05078-f001:**
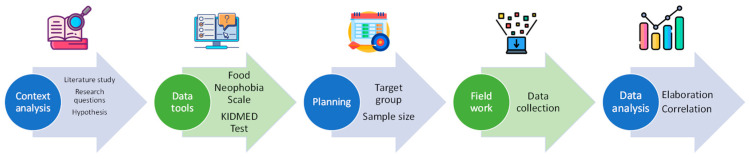
Study phase description in sequential order.

**Figure 2 nutrients-15-05078-f002:**
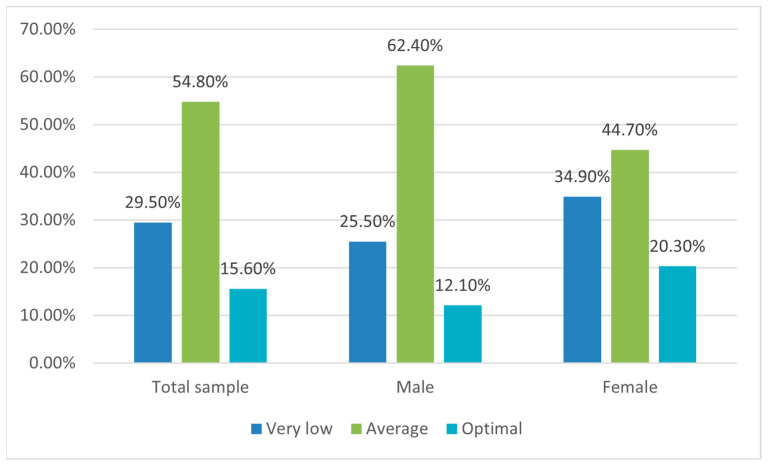
Adherence to the Mediterranean Diet (AMD) in the total sample for males and females.

**Figure 3 nutrients-15-05078-f003:**
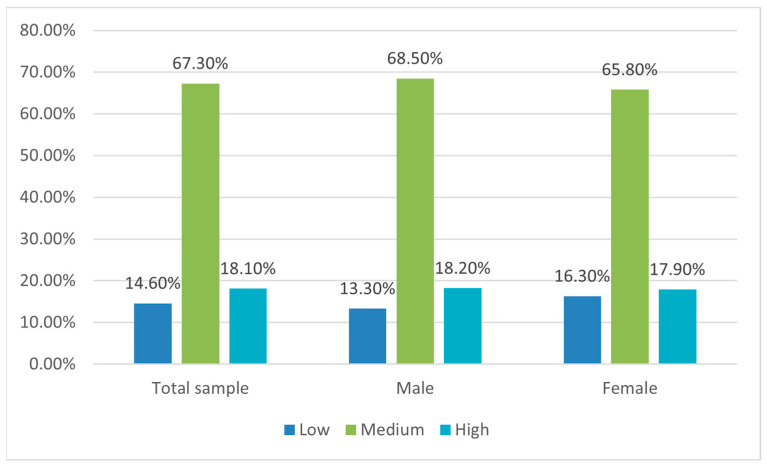
Level of Food Neophobia (FN) in the total sample for both males and females.

**Figure 4 nutrients-15-05078-f004:**
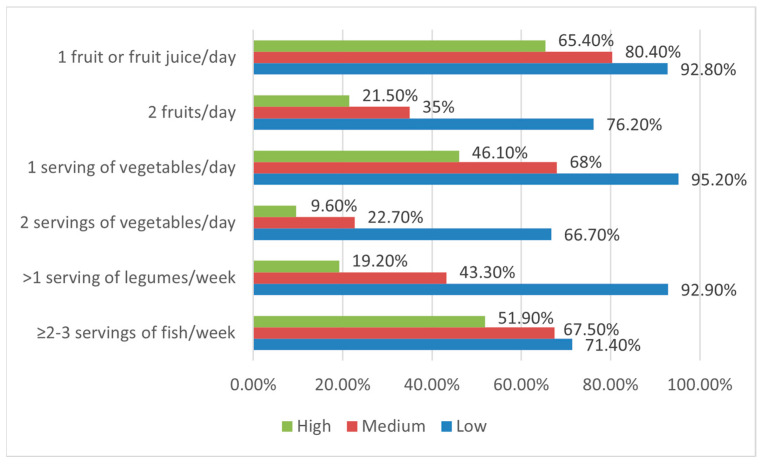
Fruit, vegetables, legumes, and fish consumption at different Food Neophobia (FN) levels.

**Table 1 nutrients-15-05078-t001:** Characteristics of the sample (n = 288).

	N	(%)
**Gender**		
Females	123	42.7
Males	165	57.3
**Age**		
3–5 years	83	28.8
6–11 years	205	71.2
**Ethnic group**		
African	13	4.5
Asiatic	4	1.4
Caucasian	229	79.8
Eurasian	37	12.9
Hispanic	4	1.4
**Parents employment**		
Both parents employed	253	87.8
≤1 parent employed	35	12.2
**Household income**		
Up to 10,000 euros	26	9
Between 10,001 and 25,000 euros	91	31.6
Between 25,001 and 40,000 euros	103	35.8
Between 40,001 and more	68	23.6
**Children per family**		
1	108	37.5
2	145	50.3
≥2	35	12.2
**Body Mass Index (BMI)**		
Underweight	16	5.6
Normal weight	129	44.8
Overweight	60	20.8
Obesity	83	28.8

**Table 2 nutrients-15-05078-t002:** Association between sociodemographic characteristics of the sample, Body Mass Index (BMI) categories according to [56,57], and Adherence to the Mediterranean Diet (AMD) resulting from the KIDMED test * *p* < 0.05.

AMD Levels		Very Low (≤3)		Average (4–7)		Optimal (≥8)	
		n	%	n	%	n	%
Sociodemographic characteristics and BMI							
Gender	Males	42	49.4	103 *	65.2	20	44.4
	Females	43	50.6	55	34.8	25	55.5
Household income	<10,000 EUR	9	10.6	11	6.9	6	13.3
	10,000–25,000 EUR	28	32.9	46	29.1	17	37.8
	25,000–40,000 EUR	30	35.3	62	39.2	11	24.4
	>40,000 EUR	18	21.2	39	24.7	11	24.4
Parents employment	Both parents employed	68	80	147 *	93	38	84.4
	≤1 parent employed	17	20	11	7	7	15.6
Children per family	1	35	41.2	65	41.1	8	17.8
	2	40	47.1	73	46.2	32 *	71.1
	>2	10	11.8	20	10.8	5	11.1
Ethnic group	African	1	1.2	9	5.7	3	6.7
	Asiatic	2	2.3	1	0.6	1	2.2
	Caucasian	69	81.2	129	82.2	31	68.9
	Eurasian	11	13	16	10.2	10	22.2
	Hispanic	2	2.3	2	1.3	0	0
BMI	Underweight	3	3.5	9	5.7	4	8.9
	Normal weight	42	49.4	68	43	19	42.2
	Overweight	9	20	40	25.3	11	24.4
	Obesity	31	36.5	41	26	11	24.4

**Table 3 nutrients-15-05078-t003:** Association between sociodemographic characteristics of the sample, Body Mass Index (BMI) categories according to [56,57], and level of Food Neophobia (FN) resulting from the Child Food Neophobia Scale (CFNS) * *p* < 0.05.

FN Levels		Low		Medium		High	
		n	%	N	%	n	%
Socio-demographic characteristics and BMI							
Gender	Males	22	52.4	113	58.2	30	57.7
	Females	20	47.6	81	41.8	22	42.3
Household income	<10,000EUR	2	4.8	19	9.8	5	9.6
	10,000–25,000 EUR	8	19	67	34.5	16	30.8
	25,000–40,000 EUR	18	42.8	68	35.1	17	32.7
	>40,000 EUR	14	33.3	40	20.6	14	26.9
Parents employment	Both parents employed	40	95.2	170	87.6	43	82.7
	≤1 parent employed	2	4.8	24	12.4	9	17.3
Children per family	1	7 *	16.7	75	38.7	26	50
	2	25	59.5	98	50.5	22	42.3
	>2	10	23.8	21	10.8	4	7.7
Ethnic group	African	2	4.8	10	5.2	1	1.9
	Asiatic	0	0	4	2.1	0	0
	Caucasian	34	80.9	153	79.3	42	80.8
	Eurasian	6	14.3	22	11.4	9	17.3
	Hispanic	0	0	4	2.1	0	0
BMI	Underweight	0	0	12	6.2	4	7.7
	Normal weight	26	61.9	81	41.8	22	42.3
	Overweight	16	38.1	36	18.5	8	15.4
	Obesity	0 *	0	65	33.5	18	34.6

**Table 4 nutrients-15-05078-t004:** Association between Adherence to the Mediterranean Diet (KIDMED total score and component scores) and Food Neophobia (FN) * *p* < 0.05.

FN Levels		Low		Medium		High	
		N	%	n	%	n	%
KIDMED total score	Very low	1	2.4	57	29.4	27 *	51.9
	Average	26	61.9	109	56.2	23	44.2
	Optimal	15	35.7	28	14.4	2	3.8
KIDMED test components							
Fruit or fruit juice every day	Yes	39	92.9	156	80.4	34	65.4
	No	3	7.1	38	19.6	18 *	34.6
Second fruit every day	Yes	32 *	76.2	67	34.5	11	21.2
	No	10	23.8	127	65.5	41	78.8
Fresh or cooked vegetables regularly once a day	Yes	40 *	95.2	132	68	24	46.2
	No	2	4.8	62	32	28	53.8
Fresh or cooked vegetables more than once a day	Yes	28 *	66.7	44	22.7	5	9.6
	No	14	33.3	150	77.3	47	90.4
Fish at least 2–3 times per week	Yes	30	71.4	131	67.5	27	51.9
	No	12	28.6	63	32.5	25 *	48.1
Fast-food more than once a week	Yes	1	2.4	40	20.6	13	25
	No	41*	97.6	154	79.4	39	75
Legumes more than once a week	Yes	39 *	92.9	84	43.3	10	19.2
	No	3	7.1	110	56.7	42	80.8
Cereals or grains (bread, etc.) for breakfast	Yes	13	31	35	18	6	11.5
	No	29	69	159	82	46	88.5
Nuts at least 2–3 times per week	Yes	7	16.7	56	28.9	10	19.2
	No	35	83.3	138	71.1	42	80.8
Skips breakfast	Yes	2	4.8	22	11.3	10	19.2
	No	40	95.2	172	88.7	42	80.8
Dairy products for breakfast (yogurt, milk, etc.)	Yes	40	95.2	180	92.8	44	84.6
	No	2	4.8	14	7.2	8	15.4
Commercially baked goods or pastries for breakfast	Yes	36	85.7	156	80.4	43	82.7
	No	6	14.3	38	19.6	9	17.3
Sweets and candy several times every day	Yes	11	26.2	79	40.7	25	48.1
	No	31	73.8	115	59.3	27	51.9

**Table 5 nutrients-15-05078-t005:** Linear model between Food Neophobia (FN) and Adherence to Mediterranean Diet (AMD).

Variable	Category	Estimation	*p*-Value
FN Continuous score	Intercept	8.61	0.000
	β	−0.09	0.000
FN levels	Low	6.93	0.000
	Medium	4.92	0.000
	High	3.54	0.000

**Table 6 nutrients-15-05078-t006:** Complete linear model between all variables (sociodemographic information and Food Neophobia score FN) and Adherence to the Mediterranean Diet (AMD).

Variable	Category	Estimation	*p*-Value
Intercept		9.85	0.000
FN continuous score	Β	−0.09	0.000
Gender	Males	0.22	0.390
Household income	10,000–25,000 EUR	−0.52	0.447
	25,000–40,000 EUR	−1.25	0.080
	>40,000 EUR	−0.82	0.269
Parents employment	Both parents employed	1.40	0.025
Children per family	2	0.42	0.143
	>2	−0.39	0.378
Ethnic group	Asiatic	−0.20	0.864
	Caucasian	−1.27	0.049
	Eurasian	−0.92	0.188
	Hispanic	−2.74	0.026
Body Mass Index (BMI)	Normal weight	−0.97	0.075
	Overweight	−0.67	0.249
	Obesity	−0.63	0.279

**Table 7 nutrients-15-05078-t007:** Complete linear model between all sociodemographic variables and Food Neophobia levels (FN) and Adherence to the Mediterranean Diet (AMD).

Variable	Category	Estimation	*p*-Value
FN levels	Low	7.10	0.000
	Medium	4.94	0.000
	High	3.69	0.000
Gender	Males	0.22	0.418
Household income	10,000–25,000 EUR	−0.79	0.150
	25,000–40,000 EUR	0.38	0.345
	>40,000 EUR	0.34	0.212
Parents employment	Both parents employed	1.43	0.032
Children per family	2	−0.10	0.759
	>2	−0.44	0.076
Ethnic group	Asiatic	−0.42	0.739
	Caucasian	−1.29	0.062
	Eurasian	−0.95	0.203
	Hispanic	−2.69	0.041
Body Mass Index (BMI)	Normal weight	−0.58	0.170
	Overweight	0.64	0.070
	Obesity	−0.34	0.207

## Data Availability

The archived data and all the elaboration and analysis generated and used for the presentation of results in this study are fully available on request from the corresponding author.

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
