# Peer review of "Adherence to Mediterranean Diet and Food Neophobia Occurrence in Children: A Study Carried out in Italy"

_nutrients, 2023, doi:10.3390/nu15245078_

Round 1

Reviewer 1 Report

Comments and Suggestions for Authors

The manuscript describes a very basic survey study on Mediterranean diet patterns in Italian children. Below are questions and suggestions for the Authors.

1. Why was the study only conducted for 3 days?

2. Some of the results were given as mean +/- SD, and the Authors did not describe in the section devoted to statistical analysis why such a decision was made, and no tests assessing the distribution of variables were used.

3. Below the tables there is no explanation of the abbreviations used, e.g. BMI

4. The Authors provide exact p-values when there is no statistical significance (e.g. lines 194-196), which I consider unnecessary.

5. The study is only a survey, so it should be enriched with interesting correlations and statistical analyses.

6.The conclusions lack descriptions of future research directions that should be initiated or expanded. In what context is there a gap in scientific knowledge that should be filled?

7. Check the references again, because some of them lack of journal abbreviations. In item 50, one of the authors is in capital letters.

8. Only ¼ of all literature items are studies from the last 5 years. The paper should be enriched with new published research.

Reviewer 2 Report

Comments and Suggestions for Authors

This study aims to clarify the association between FN and AMD in Italian children. In addition, we aim to describe the relationship between sociodemographic factors and the nutritional status of the children. Demonstrating these will allow us to consider factors that influence future health, well-being, and quality of life. Therefore, this study is of great significance.

Major Comments

The study needs to clearly explain the selection and exclusion criteria regarding the research subjects. It is essential to clarify the status of the research subjects as it provides the rationale for the study. The criteria for selecting and excluding research subjects should be clearly stated as the background for this study. Please consider this.

This study begins with the recruitment of research subjects, followed by the survey implementation and the acquisition of the survey results. During this process, did any exclusion of research subjects or results occur? If these matters have occurred, they should be indicated. Also, it would be easier to understand the flow of the research if the chronological flow of these items is shown in a flow chart. Please consider this.

In the recruitment and selection of the study subjects, it is explained that there were two groups: those for whom data was collected directly and those for whom data was collected via social media platforms (L91-95). What is the author's opinion on the possibility of bias between the two groups? Please consider.

Minor Comments

In the display of BMI results, there are four classifications: Underweight, Normal weight, Overweight, and Obesity, but the criteria for these classifications should be clearly indicated.

Round 2

Reviewer 1 Report

Comments and Suggestions for Authors

The authors responded to my comments and suggestions sufficiently.

Author Response

Thanks for the comment,

on behalf of the Authors